# A Reappraisal of Polyploidy Events in Grasses (Poaceae) in a Rapidly Changing World

**DOI:** 10.3390/biology11050636

**Published:** 2022-04-21

**Authors:** Acga Cheng, Noraikim Mohd Hanafiah, Jennifer Ann Harikrishna, Lim Phaik Eem, Niranjan Baisakh, Muhamad Shakirin Mispan

**Affiliations:** 1Faculty of Science, Institute of Biological Sciences, Universiti Malaya, Kuala Lumpur 50603, Malaysia; acgacheng@um.edu.my (A.C.); sva180026@siswa.um.edu.my (N.M.H.); jennihari@um.edu.my (J.A.H.); 2Centre for Research in Biotechnology for Agriculture (CEBAR), Universiti Malaya, Kuala Lumpur 50603, Malaysia; 3Institute of Ocean and Earth Science, Universiti Malaya, Kuala Lumpur 50603, Malaysia; phaikeem@um.edu.my; 4School of Plant, Environmental, and Soil Science, Louisiana State University Agricultural Center, Baton Rouge, LA 70803, USA

**Keywords:** climate change, evolution, food security, plant breeding, plant diversity, polyploids

## Abstract

**Simple Summary:**

Polyploidy events have long been recognized as the primary driving force behind the survival of the vast majority of plant lineages, playing critical roles in crop evolution, speciation, and domestication. The fascinating modern genomics era has revealed that the genomes of all flowering plants have been polyploidized multiple times. To help safeguard the future of the global food supply in the face of climate change, a thorough understanding of polyploid genomes is critical, especially for the improvement of members of the grass family, Poaceae, which includes the world’s big three cereals—rice, wheat, and maize—as well as some potential underutilized ancient grasses (such as teff) that are less well known or studied. This is becoming easier with rapid advances in genomic technologies. However, there are critical knowledge gaps and research needs among the well-studied and less-studied Poaceae polyploids that must be addressed, particularly the characteristics of polyploids that might either complicate or facilitate crop improvement programs. This review discusses the potential role for the Poaceae family in providing insights into the significance and specific features of polyploid genomes, as well as the emerging challenges and prospects for furthering polyploidy understanding in crop improvement.

**Abstract:**

Around 80% of megaflora species became extinct at the Cretaceous–Paleogene (K–Pg) boundary. Subsequent polyploidy events drove the survival of thousands of plant species and played a significant historical role in the development of the most successful modern cereal crops. However, current and rapid global temperature change poses an urgent threat to food crops worldwide, including the world’s big three cereals: rice, wheat, and maize, which are members of the grass family, Poaceae. Some minor cereals from the same family (such as teff) have grown in popularity in recent years, but there are important knowledge gaps regarding the similarities and differences between major and minor crops, including how polyploidy affects their biological processes under natural and (a)biotic stress conditions and thus the potential to harness polyploidization attributes for improving crop climate resilience. This review focuses on the impact of polyploidy events on the Poaceae family, which includes the world’s most important food sources, and discusses the past, present, and future of polyploidy research for major and minor crops. The increasing accessibility to genomes of grasses and their wild progenitors together with new tools and interdisciplinary research on polyploidy can support crop improvement for global food security in the face of climate change.

## 1. Introduction

The scientific community has long sought to understand how some species were able to cross the Cretaceous–Paleogene (K–Pg, or formerly K–T) boundary, surviving the catastrophic mass extinction while adapting to an ever-changing world [1,2]. Interestingly, recent paleontological research has revealed that the evolution of South American rainforests, from open canopies with dense stands of gymnosperms to closed canopies dominated by angiosperms, was strongly influenced by the asteroid’s blast [3]. Based on the analysis of thousands of fossilized plant remnants collected from the hyper-diverse tropical rainforests, it was determined how plant communities, particularly angiosperms, recovered and thrived after the mass extinction, with increased soil nutrient availability being one of the main contributing factors [3]. Most angiosperms, which account for approximately 80% of today’s flora, have duplicated their genomes at some point during their evolution. These species are termed polyploids, possessing multiple copies of the same genome [3,4]. According to Fawcett et al. [4], the ancient doubling of the genome in these species, which peaked between 60 and 70 million years ago, provided an advantage in adjusting to dramatic environmental change, which killed off the less well-endowed species.

Whole-genome duplication (WGD), also known as polyploidy in angiosperms, has been studied for over a century, a field pioneered by notable scientists such as Hugo de Vries [5]. While polyploidy is a common mode of speciation that increases species diversity and population dynamics, most polyploidy events are not successful because new polyploids are often less adapted than their diploid parent(s) and genetic consequences such as chromosomal anomalies and genomic instability can eliminate new polyploids from a population [6,7]. Nonetheless, there are numerous polyploids in the modern era, including many diploids with polyploid ancestry. Some of these ancestral polyploidy events are responsible for the key characteristics associated with the origin and diversification of major plant lineages such as angiosperms including grasses and legumes [5,8].

Although polyploidy is considered prevalent, it is important to note that not all polyploidizations will lead to major species diversification over time, or that a polyploid is more successful than its unique duplicated genome [9]. What is certain is that the establishment or survival of polyploids is not random and has been observed to coincide with major periods of mass extinction and/or climatic change around the world [10,11]. The K–Pg boundary extinction event is history, but current climate change continues to pose a threat to numerous plant species globally, including a suite of in-demand monoculture crops such as rice (*Oryza sativa*), wheat (*Triticum aestivum*), maize (*Zea mays*), sorghum (*Sorghum bicolor*), and sugarcane (*Saccharum officinarum*), all of which are members of the grass family (Poaceae). These crops suffer significant loss in production and quality from biotic and abiotic stresses, thus putting the world’s future food and nutrition security at risk [5,12,13]. Further genome duplications may give these crops an advantage in a rapidly changing world.

It is recognized that currently existing polyploids far outnumber the ancient polyploids, despite the latter being unearthed to reveal more lineages of life through advanced genomic approaches [5,14,15]. In the fascinating modern genomics era, sequence-based dissection of hundreds of genomes has shown that the genomes of all seed plants and angiosperms have been subjected to multiple rounds of WGD [16,17]. A thorough understanding of polyploid genomes and WGD is critical, particularly for crop improvement, to ensure the world’s food supply in the future. Humans rely heavily on crop species in the Poaceae family for sustenance and basic nutrition [18]. Wheat and sugarcane are common examples of complex Poaceae polyploids. Rice, a typical diploid species in Poaceae, is also an ancient polyploid derived from at least one event of WGD followed by diploidization that resulted in extensive gene loss and genome (re)organization [19]. The past decade has seen the rise in popularity of some ancient grasses that produce nutritious grains such as teff (*Eragrostis tef*) and finger millet (*Eleusine coracana*), both of which are projected to be important in shaping a sustainable food future [12,20]. Nonetheless, there are critical knowledge gaps and research needs related to both well-studied and less-studied Poaceae polyploids that must be addressed, particularly the characteristics of polyploids that can either complicate or facilitate crop improvement programs. This review discusses the potential role for the Poaceae family in providing insights into the significance and specific features of polyploid genomes, as well as the emerging challenges and prospects to further the understanding of polyploidy in crop improvement efforts.

## 2. Impacts of Polyploidy at a Glance: The Case of Polyploid Grasses

Polyploidy, which results from an organism’s or cell’s genome duplication, can have a significant impact on various levels of biological organization [21]—from genes to entire ecosystems (Figure 1). Polyploidy is generally divided into two types: autopolyploidy and allopolyploidy, which involve intraspecific WGD and interspecific hybridization events, respectively [22,23]. Autopolyploids typically have lower fertility, whereas allopolyploids have the potential for hybrid vigor (or heterosis) [24]. Much research effort has been directed toward allopolyploids to understand the consequences of polyploidy; however, relatively little is known about the outcomes of polyploidy in autopolyploids [5,16]. Genetic alterations to the basic omes, such as the genome and transcriptome (Figure 1), are almost always unavoidable in nascent polyploids, and usually only those with dynamism can survive [16,17]. Over the last three decades, it has become clear that relatively young allopolyploids may have a highly dynamic genome, undergoing “genomic shock”, where the genome experiences a variety of phenomena such as structural changes and massive reorganization to cope with disturbances [25,26]. Even though numerous studies on crops and their close relatives have revealed various genetic phenomena associated with polyploidy, there is a gap in understanding commonalities derived from shared polyploid cellular processes, particularly across different fields of study [21]. Polyploidy, as a biological force (Figure 1), is poised to improve crop fitness and diversity in the face of climate change as the impacts and consequences of polyploidy on adaptability and stress resilience in plants becomes better understood. The increase in an organism’s cellular ploidy caused by genome replication without mitosis, known as endopolyploidy, has been shown to play important roles in physiology and development via cellular, metabolic, and genetic effects [27]. This section discusses the key incidents and evolutionary consequences of polyploidization based on case studies of cereal crops in order to shed light on the role for polyploidy in crop development and survival under climate change and extreme environments.

### 2.1. Diversity and Evolutionary Dynamics

The K–Pg boundary was a pivotal period in Earth’s history, wiping out half of all species and drastically altering the planet’s environment. Polyploidy, followed by gene loss and diploidization, has long been recognized as a significant evolutionary force in plants and other organisms, facilitating diversity and rapid speciation [28,29,30]. For millions of years, events associated with whole-genome duplications have been a major contributor to the success of angiosperms. However, evidence for proposed ancient polyploidy events prior to the divergence of monocots and eudicots is ambiguous, especially in analyses of conserved gene order [3]. Many plant lineages, including monocots (such as rice) and eudicots (such as Arabidopsis), have had at least one paleo-polyploidy event. The determination of phylogenetic placements and timing of polyploidy has been the most pressing issue in plant evolution, with highly contradictory lines of evidence indicating that many polyploids and their phylogenetic positions remain unknown [30,31].

Domestication of wild plants around 10,000 years ago was thought to be the key innovation that led to human population expansion, which has since been sustained through continued selection and breeding of crops with novel desirable phenotypes. These phenotypes, which distinguish the domesticated crop from its wild ancestors, frequently form a group of improved traits known as “domestication syndrome” [32]. Seed dispersal, increased seed size, loss of seed dormancy and shattering, and fragrance are the examples of “domestication syndrome” [33]. The grass family, Poaceae, which includes all the major cereals, such as wheat, maize, and rice, provides a cohesive system for crop domestication studies, with the clade thought to have emerged around 75 million years ago, leading to cereal-specific lineages [33,34]. It is important to note that a genome duplication event shared by all grass crops occurred around 70 million years ago, prior to grass divergence [35]. Following that, certain cereals, such as wheat and maize, experienced additional lineage-specific polyploidy events, resulting in significant functional diversity for domestication and adaptation when selective fractionation occurred [36].

While the early stage studies of polyploidy were primarily based on the synthetic autopolyploid, Arabidopsis, much understanding of crop genome evolution following polyploidy has come from research on wheat, an allopolyploid [16,17]. Allopolyploidy was discovered to limit the chromosome number and the distribution of low-copy sequences in the hexaploid wheat genome, in comparison to those seen in its diploid progenitors (A, B, and D subgenomes), demonstrating a loss of homoeologous regions [36,37]. Polyploidy has facilitated many domestication-related and evolutionary traits in allopolyploid wheat over the last few decades, with notable examples being the fresh-threshing character [38] and nitrogen (N) uptake and assimilation [39,40].

Crop species in the Poaceae family, other than wheat, have been successful in surviving or thriving in a variety of environmental conditions, albeit with limitations in adaptive potential due to domestication, particularly of those with large genomes. Targeted selection and demographic bottlenecks during domestication are thought to have compromised adaptive variation, potentially reducing the extent of phenotypic parallelism observed during adaptation [33,41]. Nonetheless, there is a lack of understanding of parallel adaptation within crop species, which is critical in improving modern breeding [33]. Understanding polyploidy in non-crops or invasive species, such as weeds and Spartina, is also important for improving current agricultural management practices that cause rapid environmental change [42,43]. Even though plants in the Poaceae family are the dominant primary producers and the foundation of terrestrial ecosystems, the macroevolutionary dynamics of plant clades across the K–Pg boundary have received far less research attention than the prominent vertebrate clades [29,30].

### 2.2. Crop Improvement

In general, polyploidization in plants involves two distinct processes: genome merger and genome doubling (multiplication) [44]. Artificial, also known as synthetic polyploids, are frequently produced in plant breeding through a genome merger that facilitates hybridization, which involves interspecific crosses between different genotypes that typically occur within the same taxonomic species. While traditional selection is based on phenotypes, the union of different genotypes during hybridization allows interactions among differentiated alleles that result in a variety of effects (Figure 2), the most prominent of which is heterosis, also known as hybrid vigor and outbreeding enhancement [45,46]. Heterosis is frequently used to create a hybrid with superior characteristics such as higher yield and stress tolerance.

The potential for epistatic effects on heterosis in allopolyploids has been widely dis-cussed and is thought to have contributed to the successful domestication of many modern allopolyploid crop species. A recent review by Labro et al. [47] provides a clear overview on heterosis and its importance in crop breeding. Given that the number of interacting gene pairs increases with ploidy level, the situation in polyploids can be complicated. Dissecting the genetic mechanisms underlying the merge is an important goal in polyploid research because this knowledge may be used to enhance phenotypic novelty and heterosis in crops [16,46,47]. Nonetheless, unlike in diploids, the greater complexity of genotype and phenotype in polyploids makes it difficult for breeders to effectively link genotype to phenotype [24,48].

For nearly a century, crop agronomic traits have been improved by artificial induction of polyploidy via biological, chemical, or physical methods. This was especially true after the discovery in 1937 of colchicine, the most commonly used chemical agent. Colchicine inhibits mitotic spindle formation, resulting in a failure during anaphase disjunction and cytokinesis, with one cell having an extra or doubled chromosome set [49,50]. Homoeologous exchanges have been observed in young allopolyploids of various angiosperms. Mason and Wendel [51] reported the possible roles and effects of homoeologous chromosome recombination during the early stages of allopolyploid stabilization. While polyploidy can contribute to decreased fertility due to meiotic errors, it can also restore the fertility of a newly formed sterile hybrid through genome doubling. It is important to note that breeding new crop varieties through induced polyploidy may have some negative consequences, which are caused in part by genetic instability after polyploidization. This is perhaps the fundamental reason why evolutionary advantages of polyploidy are still being questioned and debated. Evidence implies that if early ploidy and species barriers are overcome, polyploidization can provide immediate fitness gains, paving the way for a transformed fitness landscape with a higher diversity of alleles. To sustain the benefits of polyploids over their diploid progenitors in the long run, ecological forces such as agricultural propagation or unique ecological niche availability are required [52].

For cereal crops, the first synthesized amphidiploid species is the triticale—a chromosome-doubled intergeneric hybrid obtained from the distinct cross between wheat species (*Triticum* spp., AA, AABB, or AABBDD) and rye (*Secale cereale* L., RR) (Figure 2). Various genome combinations and ploidy levels have resulted from the attempted crosses, including tetraploid AARR, hexaploid AABBRR, and octoploid AABBDDRR. Triticeae synthetic allopolyploid species, which frequently involve *Triticum* spp. and *Aegilops* spp., are regarded as excellent models for studying polyploid evolution, particularly for the wheat–rye hybrid triticale [53]. Triticale has a very complex genome compared to many other allopolyploids due to its large genome size and high ploidy level. Ploidy manipulation is a well-known and effective technique for developing novel phenotypes in cereal breeding. Nonetheless, severe seed sterility has been observed in rice since the 1930s as a barrier to autotetraploid cultivation. Recent research has identified two fertile high seed-setting autotetraploids, the Polyploid Meiosis Stability (PMeS) and Neo-Tetraploid lines, developed through tissue culture followed by chemical treatment and extensive hybridization between different autotetraploid rice followed by directional selection, respectively. Identifying the genes responsible for their high seed fertility may allow for the efficient development of autotetraploid rice varieties via marker-assisted selection (MAS) or genome editing [54]. Further, research on understanding the origin of these polyploids can set the platform for successful development and deployment of polyploid rice in future.

The majority of current knowledge about the consequences of polyploidy comes from studies involving specific crop species and their genomic mimics, which are frequently dynamically resynthesized from existing models of their wild diploid counterparts [16,30]. It is worth noting that genetic improvement for both cereal and non-cereal grasses with complex genomes can be difficult due to their high ploidy and level of genetic redundancy, which can hamper the efforts in transgene expression and the development of suitable molecular markers. Sugarcane is a well-known example of a non-cereal polyploid grass with a complex genome [55]. Although sugarcane is a model species for studying gene expression in allopolyploids, its complex genome is a barrier to genetic analysis, and the assemblies are highly fragmented, with little information available for complex repeats [55]. Because genetic introgression through hybridization is possible between wild and cultivated sugarcane due to the fertility of most hybrids and their diploid-like chromosome behavior, it is worthwhile to investigate the polyploidization of this important crop, which accounts for approximately 80% of global sugar production [55,56].

## 3. Dissecting the Polyploid Potential of Current and Future Crops in the Genomic Era

Cytological analysis of chromosomes was historically used to identify polyploid plants until recent decades when advances in genomics have enabled more efficient, large-scale research on more ancient and cryptic cases of polyploidy [9]. The rise of next-generation sequencing (NGS) in the early twenty-first century enabled the completion of hundreds of plant genome sequences as well as the annotation of millions of genes, including some polyploid species in the Poaceae family shown in Table 1. The first assembly of the hexaploid wheat (*T. aestivum*) genome was completed using the Illumina technology [57]. The DenovoMAGIC-2 assembler, which was recently used to assemble the allohexaploid wheat genome, is a one-of-a-kind assembler that has successfully reconstructed some polyploid and heterozygote plants with large genomes and large numbers of repetitive sequences [58].

With the introduction of third generation sequencing technologies (such as long-read sequencing and scaffolding) in the late 2000s, sequencing costs were further reduced with simplified preparatory methods, allowing for longer read lengths and more comparative genome analyses across multiple species. However, these technologies have a high error rate and necessitate very high-quality DNA [63]. Using the original strategy of whole genome profiling-based BAC sequencing, the first monoploid genome of a sugarcane hybrid was developed [64]. The Single Molecule Real-Time (SMRT) sequencing is one of the most widely used long-read sequencing technologies, having been used to assemble various crop genomes such as quinoa (*Chenopodium quinoa)* with a read length of about 12 Kbp [65]. Long-range scaffolding technologies can be used for contig extension or scaffolding to improve the contiguity of an assembly. Utilization of SMRT and Hi-C sequencing technologies in combination of Illumina short-read sequencing led to the development of an allele-defined genome assembly of *Sachharum spontaneum*, the wild autopolyploid progenitor of sugarcane [66]. Continuous advancements in polyploid crop genome sequencing and assembly will benefit modern plant breeding and aid researchers in studying the genotype–phenotype–environment relationship. Multiple high-quality reference genomes or a pan-genome, in combination with reference sub-genomes, are critical for identifying genomic variants associated with agronomic traits of interest and for better understanding of the genomes of important species, including both major and minor crops that play a role in sustaining global food security in the face of climate change.

Advances in crop genetics and genomics have paved the way for our understanding of polyploidy in crops. Recent research suggests that polyploidization is enhanced in many crops, resulting in profound changes in their plant growth and cell wall composition [38,67]. The development of a comprehensive phylogenetic framework of numerous plant genera, including many major crops, revealed that more polyploidization events occurred in domesticated crops than in wild counterparts, allowing for the formation and selection of important traits through the expansion of genetic materials [68]. One intriguing example is the free-threshing trait in hexaploid wheat, which is controlled by the Q locus, in which all three homoeologous genes from the A, B, and D sub-genomes contribute to the trait via different modes of functional evolution involving hyper-functionalization, sub-functionalization, and pseudogenization. The co-regulation of these genes, which are not found in diploid wheat, is required for the formation of the free-threshing trait, indicating that polyploid speciation was required for wheat domestication [38,67].

Sub-genomes in different species have been reported to differ based on their influences on domestication. For example, in maize which underwent WGD ~10 MYA, genes for some important traits in a dominant sub-genome showed more variation than genes in another sub-genome, demonstrating asymmetrical contributions of different sub-genomes [69]. More recently, Van Buren et al. [70] reported exceptional sub-genome stability in the minor cereal teff (*Eragrostis tef*), as well as a high degree of functional divergence between sub-genomes. These genomic resources will not only help accelerate breeding programs for this future cereal crop but will also provide fundamental insights into the evolution of polyploid genomes. With an increasing number of genome assemblies available for model, non-model, and crop species in various plant families, it is becoming easier to re-examine or re-interpret the reticulated histories of chromosomal evolution in different taxa with greater certainty [71]. Genome-wide comparisons of genomes with common ancestors will allow tracing of evolutionary differences (such as gene loss rates), providing a better understanding of processes of genome remodeling due to polyploidy, and the acquired knowledge can be transferred between crops [72,73].

## 4. Reappraising Polyploidy for Crop Improvement in the 21st Century: The Road Ahead

Polyploidy has played a key role in crop evolution and domestication for millions of years, increasing allelic diversity and fixing heterozygosity while generating new phenotypic variations as a result of duplicated genes that give rise to new functions [74]. There is no doubt that polyploidy has played a significant historical role in crop improvement, but moving forward, with the need for accelerated breeding strategies, there are significant gaps in the existing knowledge base that must be addressed, particularly in how polyploidy impacts biological processes and affects the genetic transmission of important agronomic traits between different crop species and their wild counterparts. Because polyploidy has been highly compartmentalized, the similarities between polyploids in terms of cellular processes at various levels of biological organization and diversity remain largely unexplored [21]. Although it is well known that polyploidy almost always increases cell size, resulting in a decreased ratio of cell surface/volume [75], the fundamental impact of these polyploidy-driven cell changes, which are frequently influenced by changing environments, is still unclear [21,76]. This uncertainty, which stems primarily from the lack of translatable communications across disciplinary boundaries, such as genomics, ecology, evolution, and agriculture, is regarded as one of the most significant challenges in developing strategies to combat climate and/or social crises, such as sustaining biodiversity and ecosystem services and improving agricultural yield in a rapidly changing world (Figure 3) [13,21]. Therefore, interdisciplinary research on polyploidy involving multiple disciplines may facilitate discovery of novel insights into the million-year-old phenomenon at various levels of organization, from cells to organisms (Figure 1).

It is worth noting that the cellular level for ploidy (or endopolyploidy, endoreplication, and endoreduplication) responses can be stimulated by ancient mutualistic associations [77]. Endopolyploidy has the potential to trade off with whole-organism ploidy, demonstrating that polyploidy can affect phenotype at both cellular and organismal levels [76,77,78]. Although stress has been shown to cause polyploidy [5,79,80], the molecular interactions between stress reduction or stress sensing and ploidy increases in (polyploid) cells and organisms remain unknown [77]. It was not until the early 2000s that scientists realized how polyploidy in certain plant tissues can lead to increased evolutionary and agronomic success. While polyploidy has brought enormous benefits to crop improvement and can aid in the resolution of some global crises (Figure 3), it is not without drawbacks. For example, in a newly formed polyploid, deleterious or negative mutations may accumulate over time and be harmful to the population at a high frequency [77,81]. Baduel et al. [52] and Spoelhof et al. [82] discussed the complicated mixture of advantages and drawbacks for polyploidy. To summarize, further research is needed to elucidate the mechanisms underlying polyploidy-induced novelty, as well as to understand various genetic phenomena associated with polyploidy in crops, which come with a set of limitations and challenges, as illustrated in Figure 3. On the plus side, advances in genetic and genomic technologies have made the genomes of many more Poaceae crops and their wild relatives accessible. Together with new tools to effectively measure ambiguous phenomena, this is opening up numerous opportunities to maximize the potential of polyploid crop species for improved global food and nutrition security under climate change (Figure 3).

## 5. Conclusions

Plant polyploidization has a delicate mix of benefits and drawbacks that have been disputed for a long time and continue to be debated. Although polyploids may initially struggle to compete with their diploid progenitors, they may be able to survive or thrive under different conditions, particularly rapidly changing environments owing to climate change. In recent years, there has been an increase in support for the idea that polyploidization can act as a buffer to mitigate the effects of both abiotic and biotic stresses. Nonetheless, the implications and extent to which polyploidy may confer a selective advantage under different stresses may vary greatly between species, necessitating further research. We believe that developing polyploid crops, particularly the major and minor cereals of the Poaceae family that feed more than half of the world’s population, could be a viable option for promoting sustainable agriculture.

## Figures and Tables

**Figure 1 biology-11-00636-f001:**
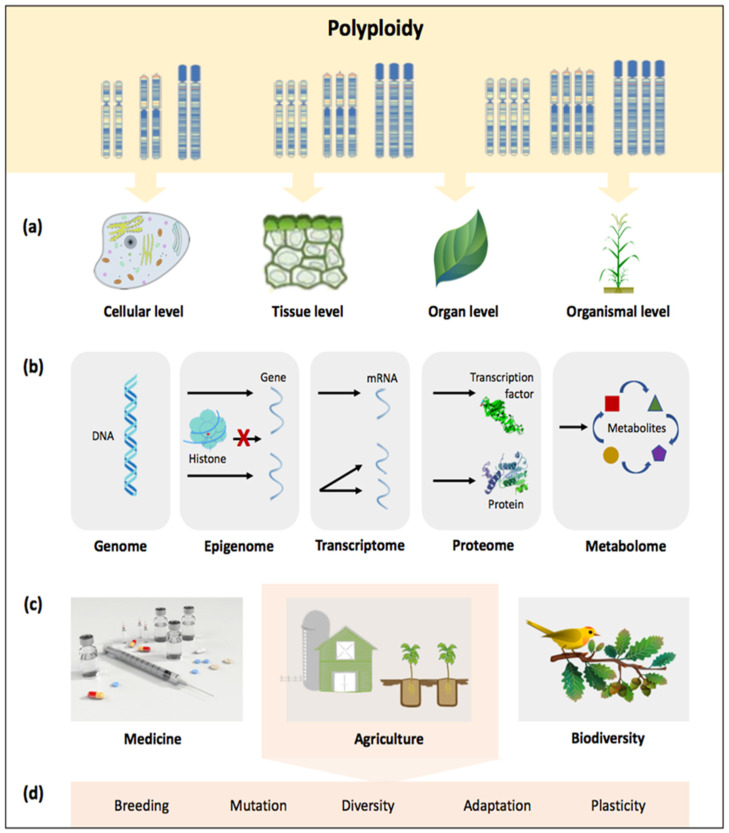
Advances in understanding polyploidy, with its role and implications from cells to ecosystems; (**a**) polyploidy involves the fusion of two or more genomes within one nucleus, affecting cells, tissues, organs, and organisms; (**b**) polyploidy as the product of a dynamic process in which restructuring of the genome, epigenome, transcriptome, proteome, and metabolome occurs through various changes; polyploidy as a biological force in (**c**) various fields and more specifically, (**d**) agriculture.

**Figure 2 biology-11-00636-f002:**
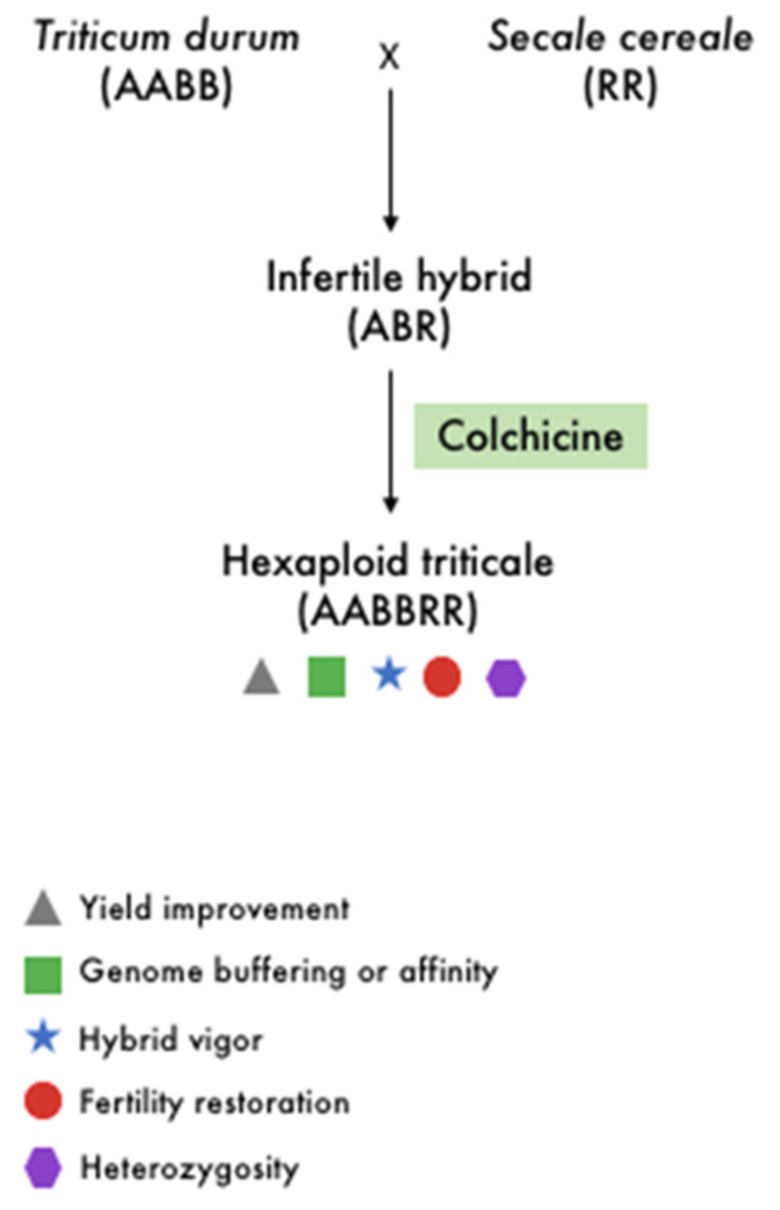
Schematic representation of artificial polyploidy of wheat with some of the main implications of polyploidy in crop improvement.

**Figure 3 biology-11-00636-f003:**
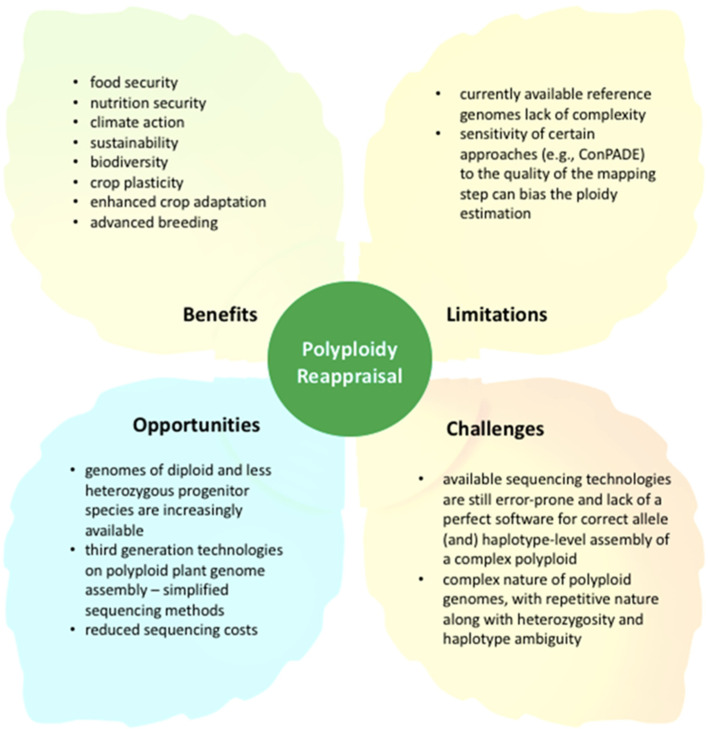
Benefits, opportunities, limitations, and challenges of reappraisal of polyploidy events in crops.

**Table 1 biology-11-00636-t001:** Sequenced polyploid genomes of grasses (Poaceae) since 2010.

Year	Crop	Genome Size (Mb)	Ploidy Level	Ploidy	Propagation	References
2010	Wheat (*Triticum aestivum*)	~15,345	Allohexaploid	6x = 42	Selfing	[57]
2014	Wild rice (*Oryza minuta*)	~450	Tetraploid	4x = 48	Selfing	Oryza Comparative Sequencing Project
2014	Teff (*Eragrostis tef*)	~607	Allotetraploid	4x = 40	Selfing	[59]
2017	Finger millet (*Eleusine coracana*)	~1196	Allotetraploid	4x = 36	Selfing	[60]
	Wild emmer wheat (*Triticum dicoccoides*)	~10,495	Tetraploid	4x = 28	Selfing	WEWseq Consortium
2018	Wild rice (*Oryza coarctata*)	~665	Tetraploid	4x = 48	Clonal	[61]
2019	Broomcorn millet (*Panicum miliaceum*)	~848	Allotetraploid	4x = 36	Selfing	[62]

## Data Availability

Not applicable.

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
