# Peer review of "A Reappraisal of Polyploidy Events in Grasses (Poaceae) in a Rapidly Changing World"

_biology, 2022, doi:10.3390/biology11050636_

Round 1
Reviewer 1 Report
The paper by Cheng et al. entitled “A reappraisal of polyploidy events in grasses (Poaceae) in a rapidly changing world” represent the review article on the impact of polyploidy in the evolution and domestication of the species of the Poaceae family. The authors stressed the importance of understanding the role and implication of polyploidy from cell to ecosystem through integration of various levels of research (genomics, proteomics, transcriptomics etc.) and suggested interdisciplinary approach for crop improvement in the sense of climate changes. They also indicate importance to include major and minor crops as a food source in the global research of polyploidy. They represent benefits and limitations of ployploids and point out opportunities and challenges in the future work. The paper is clear written and based on representative number of relevant articles in the field.
I have only one minor comment:
Line 78-79. K-T should be replaced with K-Pg
Reviewer 2 Report
Overall, I believe this is quite an interesting and valuable manuscript. It is informative and well-written. I only have a few suggestions that should be considered before publication of the paper, listed below:
- Please use alphabetical order of keywords and do not repeat words from the title
- The English form is generally good, although some punctuation errors require correction, e.g. line 21
- Please use full botanical names of species, e.g. Chenopodium quinoa Willd.
- Line 287: unclear
- I suggest providing a Conclusions chapter at the end of the text.
After incorporating all the necessary changes, the ms can be accepted for publication.
Reviewer 3 Report
In this paper, the authors describe polyploidy through the history of angiosperms and the role of grasses as major crops. They relate polyploidy as a trait that could enhance yield in grasses to substantiate the global food supply.
The paper is interesting overall but feels quite surface-level, in that not much is described in detail. I come away from the paper without an explanation for how heterosis works, why it benefits polyploids, how polyploidy contributes to plant traits generally and to agronomically valuable traits specifically, why grasses in particular are a good target for improvement via polyploidy, etc. The paper doesn’t present any new data, compile existing data in an interesting way, or reanalyze data to make broader conclusions.
Below are additional points that may help expand the paper.
I think polyploidy is interesting outside of an agricultural context, and so appreciate the focus on food crops but find it limiting. Polyploidy in grasses is worth studying on its own in my opinion.
The authors state that the origin of polyploidy in a population often comes at a cost and is outcompeted by diploid individuals, such that polyploidy isn’t sustained longterm. Yet the authors later state (and there are plenty of studies that also say) that polyploidy is beneficial for stress tolerance and other traits relative to diploidy, and thus can help sustain yields (e.g. as food crops) in changing world. Clarification can help because in one context, diploids are better than polyploids, and the reverse is true in another context. I don’t see these contexts adequately described.
Since polyploidy is part of the evolutionary history of all angiosperms, I would guess that a very early polyploidy even occurred, or more likely, independent polyploidy events are spread throughout the angiosperm lineage through time. It is not clear to me that any major polyploidy events occurred before/around the KT event, and so am left wondering to what degree the survival of plant lineages through KT is at all related to polyploidy.
I found figure 1 to be unhelpful and actually a bit confusing until I read the figure legend. Even then, I found its utility limited.
These topics could be elaborated: interaction of ploidy and the environment, how ploidy relates to changing conditions, the degree to which food production of grasses is determined by traits affected by ploidy, how much of the world’s food supply on these crops in the context of a changing world, why ancestral types like teff could be useful as food crops and how that relates to their ploidy, etc.
Wendel has some key papers on how polyploidy works, including how the two (or more) genomes “get along” within the same cell and their effect on agronomic traits. Scholes has a paper on endopolyploidy as a stress response and another showing the tradeoff between polyploidy and endopolyploidy with regard to stress. Some of these key papers felt missing since they are so relevant and can help elaborate these key areas of the paper.
Round 2
Reviewer 3 Report
I thank the authors very much for their thoughtful responses to my suggestions and questions. I appreciate that many of the topics I mentioned are reviewed extensively elsewhere and so mentioning these papers explicitly helps the reader access related information if they choose. It is understandable therefore to not elaborate all of these topics, and I like the extent to which the authors now point the reader the new references. I am also ok with Figure 1 if the authors deem it critical/important.